# Exploring Fully Biobased Adhesives: Sustainable Kraft Lignin and 5-HMF Adhesive for Particleboards

**DOI:** 10.3390/polym15122668

**Published:** 2023-06-13

**Authors:** Liam Dorn, Arthur Thirion, Masoumeh Ghorbani, Luis M. Olaechea, Ingo Mayer

**Affiliations:** Institute for Materials and Wood Technology, Bern University of Applied Sciences (BFH), CH-2500 Biel, Switzerland; dornl1@bfh.ch (L.D.); tia2@bfh.ch (A.T.); ohl1@bfh.ch (L.M.O.)

**Keywords:** bio-based adhesive, Kraft lignin, 5-HMF, wood-based panels, particleboards, wood-based composites

## Abstract

Most adhesives used in the wood-based panel (WBP) industry are petroleum-based and are associated with environmental impact and price fluctuations. Furthermore, most have potential adverse health impacts, such as formaldehyde emissions. This has led to interest from the WBP industry in developing adhesives with bio-based and/or non-hazardous components. This research focuses on the replacement of phenol-formaldehyde resins by Kraft lignin for phenol substitution and 5-hydroxymethylfurfural (5-HMF) for formaldehyde substitution. Resin development and optimization was carried out regarding varying parameters such as molar ratio, temperature or pH. The adhesive properties were analyzed using a rheometer, gel timer and a differential scanning calorimeter (DSC). The bonding performances were evaluated using an Automated Bonding Evaluation System (ABES). Particleboards were produced using a hot press, and their internal bond strength (IB) was evaluated according to SN EN 319. Hardening of the adhesive could be achieved at low temperatures by increasing or decreasing the pH. The most promising results were obtained at pH 13.7. The adhesive performances were improved by adding filler and extender (up to 28.6% based on dry resin) and several boards were produced reaching P1 requirements. A particleboard achieved a mean IB of 0.29 N/mm^2^, almost reaching almost P2 requirements. However, adhesive reactivity and strength must be improved for industrial use.

## 1. Introduction

Globally in 2021, around 396 million m^3^ of wood-based panels (WBP) were produced [1]. In order of decreasing total global production volume, the main types are oriented strand boards, plywood, medium/high-density fiberboard, particleboards and oriented strand boards [1]. WBP consists mainly of wood with 2–14% adhesive content (dry resin/dry wood mass) [2].

WBPs are mainly produced through hot pressing, where a mixture of wood, adhesive, and additives such as wax and emulsifiers are pressed at a specific pressing pressure of 2 to 4 N/mm^2^ at temperatures spanning from 200 to 220 °C [2,3]. The adhesives used in the manufacturing of WBP include urea-formaldehyde (UF), phenol-formaldehyde (PF), melamine-formaldehyde (MF), melamine-urea-formaldehyde (MUF), and resorcinol-formaldehyde (RF), but also polyurethane adhesives based on MDI [4,5]. Among these adhesives, formaldehyde-based adhesives (UF, PF, MF, MUF and RF) represent approximately 95% of the adhesives used for WBP, with UF being the most widely used, representing 85% of the total production volume worldwide [5]. Formaldehyde-based adhesives have drawbacks, as indoor formaldehyde emissions are associated with a range of adverse impacts on human health, including throat, nose and eye irritation [6].

Moreover, formaldehyde is classified as a 1B carcinogenic, according to the European Parliament and the European Council 605/2014 [7]. Therefore, strict indoor emissions standards, including EN 13986 [8], have pushed the market to reduce the amount of formaldehyde emitted by wood-based panels. These formaldehyde-based adhesives are also produced using petroleum-based compounds associated with higher environmental impacts than certain bio-based alternatives [9] and lead to dependence on crude oil supply and its associated price volatility. Within this group of formaldehyde-based adhesives, there are differing properties. UF is low cost, rapid curing and possesses good dry bond strength but has low water resistance, restricting it to indoor use [10]. This means that UF resins are modified via partial or full urea replacement for high-humidity applications of more expensive compounds, such as melamine, phenol or resorcinol [11,12]. PF resins are the preferred adhesives for producing exterior-grade panels and represent the second most consumed type of wood adhesive after UF [13]. As an alternative to UF resins, polyurethanes can also be used, as they possess higher dimensional stability and increased moisture resistance compared to UF [12]. Usually, such polyurethanes are based on methylene diphenyl diisocyanate (MDI) and display a better VOC profile [12] but they remain a hazard at the production site and require a high level of health protection [14]. Moreover, it remains petroleum-based and suffers from technical issues due to its propensity to generate sticky build-up on the press surface and its slower curing time when compared to formaldehyde-based resins [15]. The issues with the current adhesive systems described above have led to much interest in the wood industry regarding substituting formaldehyde and other fossil-derived components in WBP adhesives. In this body of work, we concentrate on phenol and formaldehyde substitution in PF resins.

Lignin has excellent potential for phenol substitution due to its phenolic structure, and low price [16]. Technical lignin refers to Lignin that is isolated from by-streams in lignocellulosic refiners. These included Kraft, soda, organosolv and hydrolysis Lignin and lignosulphonates [17]. These have different properties, including molecular weight, polydispersity, homogeneity and the presence of certain functional groups [17]. Currently, Kraft and lignosulfonate represent the largest by volume, making them the most interesting for high-volume applications, such as in adhesives for wood-based panels. Especially, Kraft lignin, with a high availability potential and further improvement in extraction technology [18], is a suitable candidate to replace such large volume resin as PF resin, with a worldwide production estimated at 5 mio tons/year. Kraft and lignosulfonate are associated with high ash content compared to the other technical lignin, and both have sulfur present. Kraft exhibits between 1 and 3% sulfur content in H-bonds compared to 3.5–8% in lignosulfonate [17]. Sulfur is associated with odor problems and can inhibit catalysts [17]. Kraft lignin can be extracted from the black liquor of the Kraft pulping processes; thus, the valorization of this by-product is currently of great interest to the pulp and paper sector.

On a laboratory scale, 100% of phenol replacement could be achieved without a loss of mechanical properties when compared to PF references [19]. On an industrial scale, currently, only 50–80 wt% phenol substitution by Lignin could be achieved by BioBond using UPM BioPiva Kraft lignin [20]. This limitation in substitution rate is partially due to steric hindrance, reducing lignin reactivity with formaldehyde [21]. Lignin reaction with formaldehyde is a two-step process. Firstly, there is activation by methyl-olation at the C5 position of the guaiacol units in an alkaline medium (Figure 1A), followed by a condensation reaction of the methylal groups previously formed [21]. In addition to the potential of Lignin as phenol replacement, the non-toxic 5-hydroxymethylfurfural (5-HMF) has potential as a formaldehyde replacement (Figure 1B). Moreover, 5-HMF can be produced from biomass containing fructose or glucose. This is possible via the dehydration of fructose under acidic conditions. For glucose, a previous isomerization into fructose is required [22]. From a chemical point of view, 5-HMF has the advantage of bearing on its furan ring, in addition to the aldehyde group, a hydroxy-methylene group that can contribute to the hardening of the resin via the formation of methylene bridges.

A phenol 5-HMF adhesive has been previously produced on a laboratory scale for use in fiber-reinforced composites. This adhesive was synthesized by Zhang et al. using sawdust bio-oil as phenol replacement and glucose for in situ 5-HMF formation, which would replace formaldehyde [24]. However, full replacement of phenol by sawdust bio-oil showed a decrease in the resulting molecular weights, indicating limitations in the polycondensation reaction. In this work, we aim to completely replace phenol, formaldehyde or any other petroleum-based components.

## 2. Materials and Methods

### 2.1. Materials

Crystalline 5-Hydroxymethylfurfural (95 wt%) and in solution (aqueous 22 wt%) were purchased from AVA Biochem AG (Zug, Switzerland).

Kraft Lignin was obtained from two European providers. Source 1 was UPM BioPiva™ 395 (UPM Biochemicals, Helsinki, Finnland), with an average molecular weight of 6000 g/mol. Source 2 was LineoTMClassic (Stora Enso, Kotka, Finnland), with an average molecular weight of 6000–7000 g/mol.

Sodium hydroxide (aq. 50 wt%), and Calcium carbonate powder (≥99%) were sourced from Sigma-Aldrich Chemie GmbH (Buchs, Switzerland). Wheat flour was purchased from a local store in Switzerland. Beech veneers were supplied by Atlas Holz AG (Trübbach, Switzerland). The particles were of core-layer dimension, consisted mainly of Norway spruce (*Picea abies*), with smaller portions of recycled material, and were provided by Rauch Spanplattenwerke GmbH (Markt Bibart, Germany).

### 2.2. Resin Preparation

The resin was synthesized by adding dry kraft lignin (lignin S1 and S2) (0.05 mol C9 units) portion-wise to a solution of distilled water and sodium hydroxide (aq. 50 wt%) and maintained at 90 °C under stirring. As soon as all lignin was dissolved, a 5-HMF solution (aq. 50 wt%) was added under reflux conditions via an SPL syringe pump (World Precision Instruments Inc., Sarasota, FL, USA). After addition, the mixture was maintained at 90 °C for a further 60 min prior to being cooled to room temperature via the use of an ice bath. The molar ratio of the adhesive used in board production was 2:1 5-HMF to Lignin, 0.8:1 NaOH to Lignin, and the amount of water was 1.25:1 relative to the dry mass of Lignin. A step-by-step synthesis protocol can be found in Appendix A. Solid content was measured using a Rotavapor R-124 rotary evaporator (BUCHI Corporation, Denver, CO, USA) with a water temperature of 50 °C and pressure was progressively reduced from atmospheric pressure to 0.035 bar until the mass remained stable.

### 2.3. Viscosity Evaluation

Viscosity was measured at 20 °C and 1 bar using a viscometer DV2T from Brookfield, Middleboro, MA, USA, in a conditioned environment of 20 ± 2 °C and 1.01 bar (1 atm).

### 2.4. Filler and Extender Content

The filler and extender used were calcium carbonate and wheat flour, respectively, always added in a 1:1.5 ratio (calcium carbonate/wheat flour) based on weight. The amount of both in relation to dry resin mass was varied during adhesive optimization and particleboard production and testing and is discussed in the corresponding section.

### 2.5. Particleboard Preparation

The particleboard production was executed at BFH and consisted of the following steps: drying and resination of the particles, hot pressing, sample preparation and testing. Particle drying was carried out using a custom dryer built by Ruefli AG (Biel/Bienne, Switzerland). The adsorption dehumidifier CR100 from COTES A/S (Aarhus, Denmark) was run at maximum capacity with a nominal dry air flow of 100 m^3^/h and 0.94 kW heating capacity until a desired moisture content of 2–2.5% was reached. After drying, the particles were transferred to a bag and sealed for at least one week to ensure a homogenous moisture content level.

Particle blending was carried out using a 300-litre blender from Gebrüder Lödige Maschinenbau GmbH (Paderborn, Germany). The particles were first added, and then the adhesive mixture was pumped through a 6.5 mm hose, then sprayed via an air-mix nozzle with an outlet diameter of 1.8 mm and a pressure of 2 bar. A blender rotational speed of approximately 120 min^−1^ was used.

Single-layer particle boards were pressed using a HLOP 210 provided by Höfer Presstechnik GmbH (Taiskirchen, Austria) controlled via the Pressworks software. The particleboard production process was always conducted using the 5-HMF/lignin S1 adhesives prepared as described in Section 2.2 with a pH of 13.7. Production was carried out with different variations in process parameters, such as press factor, moisture content or press closing time, and variation in resin and filler/extender content. Please refer to Appendix B for further details. The following parameters where constant among all boards produced: press plate temperature of 200 °C, final board thickness of 16 mm and target density of 650 kg/m^3^. Further processing occurred once the boards had cooled overnight. The wood to adhesive dry weight ratio was kept constant at 10 for all boards except one (Board ID 14).

### 2.6. Thermal Characterization of the Resin

The curing temperature was investigated using a differential scanning calorimeter (DSC) from PerkinElmer, DSC 6000, Waltham (M.A.), USA. DSC measurements were carried out using reusable high-pressure stainless-steel pans. Heating scans were carried out at a rate of 10.00 °C/min with the temperature range depending on the mixture’s pH, investigated to avoid corrosion of the crucibles. These were as follows: 30–250 °C for pH >3, 30–200 °C for pH 2–3, 30–180 °C for pH 1.8–2, and 30–160 °C for pH < 1.8.

Gel time measurements were done using a Gel timer SLIM LINE GT-S with control unit PTC-3 from GEL INSTRUMENTE AG, Oberuzwil, Switzerland. Testing was performed using 5 g specimens in the test tube.

### 2.7. Bonding Evaluation

The adhesives’ bonding strength development during pressing was determined in accordance with ASTM D7998−15 using an Automated Bonding Evaluation System (ABES) device (Adhesive Evaluation Systems, Inc., Corvallis, OR, USA). Two beech veneer strips of 117 mm × 20 mm × 0.5 mm were glued together with a 5 mm overlap length (100 mm^2^ glue area). Therefore, a glue load of 70 g/m^2^ dry resin was applied. Then, the veneers were pressed at the desired temperature and for varying durations, cooled with a set pressure of 20 bar for 10 s and pulled apart under tensile loading. Five repetitions were performed for each pressing time at a defined temperature.

### 2.8. Moisture Evaluation

The moisture contents of the veneers and particles were measured using the Moisture Analyzer HE73 by Mettler-Toledo International Inc. (Columbus, OH, USA). The moisture content was derived from the weight difference between moist and absolute dry mass.

### 2.9. Particleboard Evaluation

IB samples were cut using a circular saw and conditioned in a standard climate room (20 °C/80% R.H.) for at least one week. IB testing was carried out according to SN EN 319 using a universal material testing machine of type Z020 equipped with an Xforce K load cell, from ZwickRoell GmbH & Co. K.G. (Ulm, Germany).

## 3. Results and Discussion

### 3.1. Adhesive Development

Firstly, the Kraft lignin from two different sources was characterized via ^13^C-NMR and yielded the following aliphatic and phenolic content (Table 1). From this, we see that both types of Kraft lignin have similar phenolic and aliphatic content levels. The former is more advantageous regarding reaction with 5-HMF. This work was focused on Source 1 (S1). Source 2 (S2) was exclusively used to evaluate the impact that would carry a change in the provider/source of Kraft lignin after development of a formulation based on source 1.

To develop a 5-HMF/lignin S1 adhesive, a range of synthesis parameters were varied: cooking time, pH, 5-HMF to lignin S1 molar ratio and Kraft lignin supplier. The starting synthesis parameters consisted of a mixture of lignin S1 and sodium hydroxide (pH = 10) in a molar ratio of 1:0.6. The molar ratio is based on the molar weight of the C_9_ units of 180 g/mol. First, to investigate a possible self-polymerization of the kraft lignin or the 5-HMF, a mixture containing lignin and another containing solely 5-HMF were cooked for 80 min at 90 °C and at pH 10. In both cases, the mixtures exhibited low levels of viscosity, thus suggesting that no self-polymerization took place or at a least at very low extents.

Then, with the addition of 5-HMF (0.5 mL/min) into the lignin S1/sodium hydroxide mixture in a 1:1 5-HMF to lignin S1 ratio, cooking time was extended from 1 h to 1.5 h and 2 h and the viscosity was monitored (Figure 2A). Contrary to the self-polymerization experiments, the viscosity increased significantly during the reaction. After 1.5 h, the viscosity reached values that begin not to be suitable for wood-based panel production. Consequently, the reaction time was fixed at 1 h. Based on this, a range of 5-HMF to lignin S1 molar ratios were studied. This time, it was observed that the viscosity of the resin increased concomitantly with the 5-HMF/lignin S1 molar ratio until a ratio of 2 was achieved (Figure 2B). However, at a 5-HMF/lignin S1 molar ratio of 3, the viscosity decreased to values similar to those obtained with a molar ratio of 0.7. At a molar ratio of 3, the gelation time also increased to 42 min, compared to 16 min at the molar ratio of a 0.7. This could be due to the increased amount of 5-HMF leading to a reduction in pH from 10.8 at 0.7 to 9.6 at 3 molar ratios (Table 2). Based on those results and considering economic factors, the molar ratio between 5-HMF and lignin S1 was maintained between 1 and 2, as 5-HMF continues to be relatively expensive.

In addition to the initial characterization, a series of ABES tests were conducted to investigate the impact of 5-HMF/lignin S1 molar ratio on the resulting tensile shear strength. The ABES results showed that the 5-HMF to lignin ratio had no impact on bond development (Figure 3A). Moreover, the tensile shear strength was quite low. To optimize the resin formulation in terms of performance, wheat flour was tested as potential extender, known in the WBP industry for its low cost and biobased character [26]. Therefore, ABES testing was performed after the addition of 10 wt% of wheat flour based on the solid content (Figure 3B). As we can see in Figure 3B, the tensile shear strength increases significantly upon addition of wheat flour. To conclude this ABES series, the lignin S1 was exchanged for S2 in the preparation of the resin and ABES was measured under the same conditions. Interestingly, we observed an improvement in the tensile shear strength when compared to 5-HMF/lignin S1 (Figure 3C). On the other hand, neither 5-HMF/lignin S2 or 5-HMF/lignin S2 with 10 wt% of extender (Figure 3D) performed better than 5-HMF/lignin S1 with 10 wt% of extender.

Finally, DSC measurements of the resins prepared using 1 and 2 molar ratios of 5-HMF/lignin S1 were performed and revealed a decrease in the curing temperature, when increasing the molar ratio (Appendix C). Indeed, the maxima decreased slightly when increasing the 5-HMF/lignin S1 molar ratio (i.e., from 183 to 178 °C). This decrease is similar independently of the source. Based on the results obtained so far, it was decided to continue using lignin S1 solely.

### 3.2. Adhesive Optimisation

To enhance the tensile performances of the resin formulation, further optimization was carried out focusing on the curing process at different pH. To evaluate the impact of pH on the curing process, a series of ABES were performed. The pH was adjusted by adding either p-toluene-sulfonic acid (pTSA) or sodium hydroxide (NaOH) to the prepared resin. The results revealed that low curing temperatures (i.e., below 100 °C) were obtained under very acidic conditions (i.e., with a pH below 1.3) (Figure 4A). In a similar trend, the curing temperature can also be decreased by increasing the pH, reaching the lowest values (i.e., 115 °C) at a pH near 13.8 (Figure 4B). A summary of the curing temperatures related to the pH is illustrated in Figure 4C. When comparing the resulting tensile shear strengths at equivalent press time, solid content and temperature in the ABES tests, it was found that alkaline conditions exhibited higher tensile shear strengths (Figure 4D). The DSC curves in alkaline conditions displayed a distinct peak, indicating a certain degree of curing. In contrast, the acidic curves appeared relatively flat, as the exotherm spanned over a large range of temperatures. Furthermore, it was observed that the resin tends to aggregate under acidic conditions, leading to an unprocessable heterogeneous mixture, probably because of the decrease in solubility of lignin under acidic conditions [18]. Considering the necessity for low curing temperatures (i.e., 90–110 °C) in WBP production, a pH of 13.7 was defined for further investigations and for the panel manufacturing process.

Further optimization was carried out on the adhesive under alkaline conditions (pH 13.7) by means of ABES, and the relationship between bonding strength, resin, filler and extender solid content were studied (Figure 5). Furthermore, calcium carbonate was added to the formulation as a filler. It has been shown that this can improve important chemical and physical characteristics in the wood-panel based industry, such as bending strength, decrease of water absorption or thickness swelling [27]. For this series of ABES, an experimentation space was defined where the total solid content was kept constant (i.e., either 45 or 60 wt%) but where the mass ratios between resin, extender and filler were varied (Figure 5A). Considering the series with 45 wt% of solid content, similar, or even slightly higher, tensile shear strength could be obtained after decreasing the amount of resin content in favor of extender and filler (Figure 5B). This reduces applications cost as less expensive 5-HMF/lignin S1 is used. The same trend could be observed in the series with a total solid content 60 wt% (Figure 5C). On the other hand, after addition of extender and filler in the 60 wt% series, the viscosity increased dramatically and this application, while feasible in ABES, becomes intricate for particleboard production. In addition, even if feasible in ABES, high viscosities could prevent absorption of the adhesive by the wood. Because of this, additional ABES tests were conducted with a lower solid content (i.e., 52.7 wt% of solid content) (Figure 5D). Interestingly, the tensile shear strengths displayed was above the 45 and 60 series. Moreover, wood failure could already be observed in some veneers after a pressing time of 30 s. This was never observed previously in our experiments. This suggests that the bond strength becomes closer to the strength of the veneer itself. Thus, it was chosen for initial particleboard production.

### 3.3. Particleboard Production and Testing

To trial the adhesive for particleboard production, a total of 21 combinations (Appendix B) of parameters were tested, resulting in 14 self-standing boards. Their corresponding internal bond (IB) is summarized in Figure 6. From the results, it was observed that the press factor had the most significant influence, as no self-standing board could be produced when the press factor was below 30 s/mm. Moreover, the IB could be increased significantly by increasing the press factor. For instance, increasing the press factor from 30 (sample #12) to 45 s/mm (sample #13) resulted in an increased IB from 0.05 to 0.27 N/mm^2^. A similar trend was observed between sample #15 and 16, where the press factor was increased from 30 to 37.5 s/mm. Interestingly, the highest IB recorded with a press factor of 30 s/mm was achieved when the pressing was performed the day after (i.e., sample #3) rather than immediately after spraying the resin onto the wood particles (i.e., sample #2). To enhance the heat transfer within the boards during the pressing process, different amounts of water were sprayed on the hot plates before pressing (i.e., sample #17 and 18) and compared to a board (i.e., sample #16) when no water was sprayed. This water spraying on the hot plates is performed to generate additional steam that should help with heat transfer. Unfortunately, this factory trick did not work, as all samples displayed similar IB (i.e., between 0.2 and 0.25 N/mm^2^). So far, the best-performing boards (i.e., sample # 3 and 13) displayed IB sufficient for P1 classification according to SN EN 312. In an attempt to reach values related to P2 classification, two panels were produced with the same wood to a dry adhesive mass ratio of 10, but where the percentage of extender and filler was increased, keeping the resin solid content constant (i.e., sample #20 and 21). Interestingly, board #20, which contained less additives (i.e., extender and filler) than #21 but more than #19, exhibited the highest IB and reached values close to the limit between P1 and P2 (0.35 N/mm^2^) classification. However, further testing is required to ensure reproducibility and define a significant 5% value.

It was also noticed during board manufacturing that moisture content seemed to affect the IB of the particle boards. To investigate this, a series of ABES was performed varying the moisture content and a negative correlation could be observed (Appendix D) between tensile shear strength and moisture content, but additional experiments should be performed to clarify this. Water could act as a plasticizer lowering the IB, suggesting that water resistance and or degree of crosslinking still need to be improved in this system in order to be used under humid conditions.

## 4. Conclusions

A fully bio-based adhesive based Kraft lignin and 5-HMF has been formulated and evaluated and applied in the production of particleboards at a lab scale. The adhesive was developed via varying synthesis parameters, such as lignin types, pH, cooking time and 5-HMF to lignin ratio. The adhesive reactivity could be tuned by varying the pH, reaching the requirement for hot pressing particle boards production under alkaline conditions. To enhance the adhesive performance, formulation modifications were made, incorporating cost-effective and bio-based additives, such as wheat flour. It could also be shown that, in addition to the actual adhesive formulation, the setting of the adhesive (e.g., water content) and other process parameters during panel manufacture have a major influence on the IB of the particle boards. Utilizing this fully biobased resin, particleboards reaching P1 classification, and nearly P2 classification, were successfully produced. However, further improvement of the reactivity of the resin is certainly required as the current press factor results in non-acceptable press times for current industrial applications. The development of this fully bio-based adhesive represents an advancement in sustainable adhesive technologies, demonstrating the potential to replace petroleum-based adhesives in particleboard production and allowing the production of boards based fully on naturally-derived raw materials.

## 5. Patents

The Patent P25109CH00—Adhesive formulation for wood panels, resulted from the research described in this paper.

## Figures and Tables

**Figure 1 polymers-15-02668-f001:**
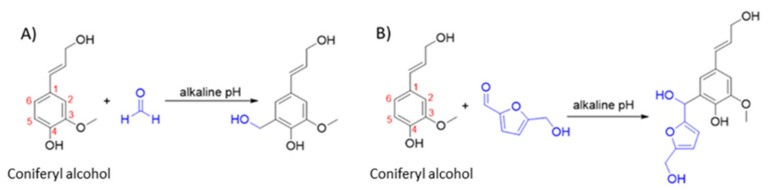
(**A**) Scheme of the electrophilic aromatic substitutions of coniferyl alcohol with formaldehyde. (**B**) Scheme of the proposed electrophilic aromatic substitutions of coniferyl alcohol with 5-HMF. This is based on information from literature [23]. The figure was drawn by one of this paper authors and coniferyl alcohol is used here solely to depict the Guaiacol unit of lignin.

**Figure 2 polymers-15-02668-f002:**
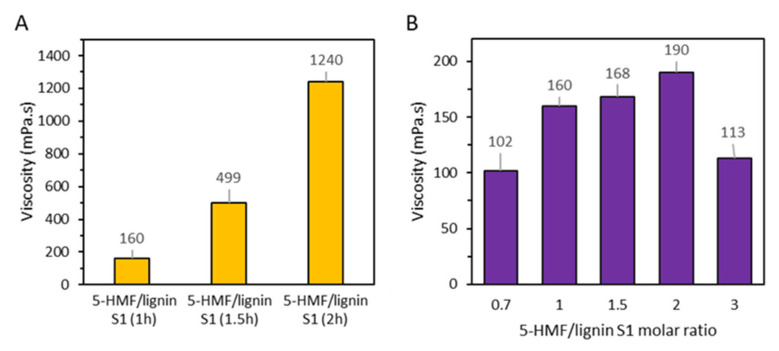
(**A**) Viscosity measured at room temperature (25 °C) after 1 h, 1.5 h and 2 h of cooking at 90 °C (**B**) Viscosity at room temperature of the adhesive with varying 5-HMF to lignin S1 molar ratio.

**Figure 3 polymers-15-02668-f003:**
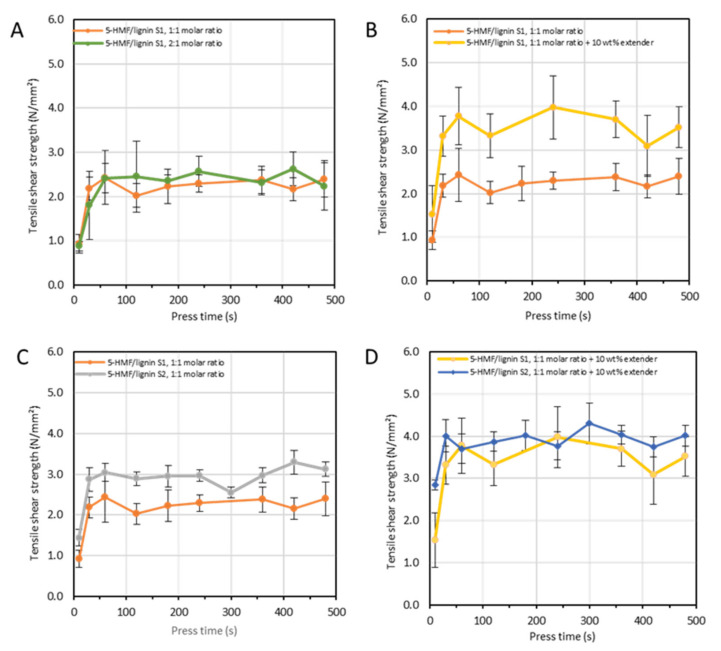
Adhesive strength development at 130 °C of different adhesive formulations. (**A**) Different 5-HMF/lignin S1 molar ratio, (**B**) with and without wheat flour as extender, (**C**) source 1 and source 2, (**D**) Different supplier with filler.

**Figure 4 polymers-15-02668-f004:**
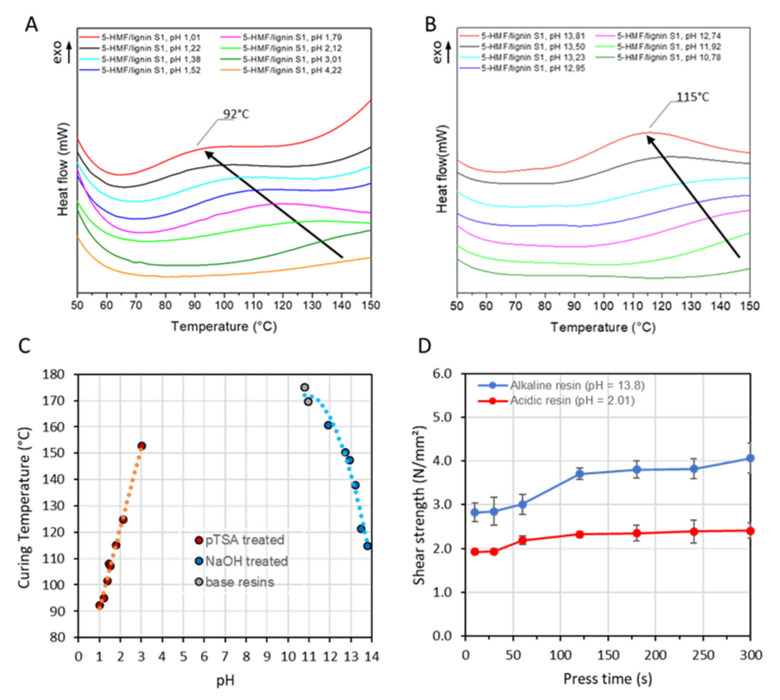
Stacking of DSC chromatographs of a 5-HMF/lignin S1 resin in (**A**) acidic conditions using p-toluene sulfonic acid to correct the pH and (**B**) under alkaline conditions using sodium hydroxide to correct the pH. (**C**) Relationship between pH and curing temperature. (**D**) Tensile shear strength after pressing a 130 °C at pH 13.8 (blue curve) and pH 2.01 (red curve).

**Figure 5 polymers-15-02668-f005:**
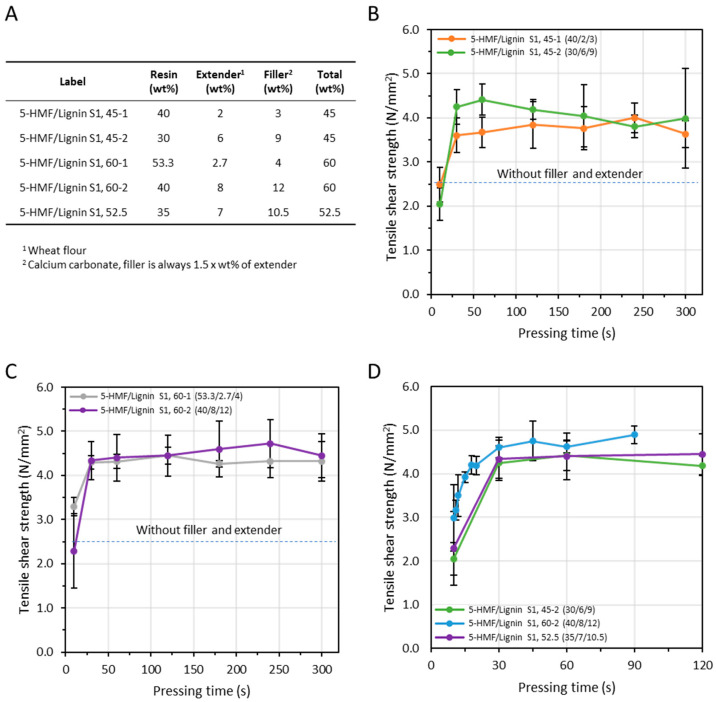
Impact of different adhesive solid content and filler and extender amount (based on dry resin) on adhesive strength development. (**A**) The formulations tested and their impact on adhesive strength development with a (**B**) total solid content of 45 wt%, (**C**) total solid content of 60 wt% and (**D**) differing total solid content.

**Figure 6 polymers-15-02668-f006:**
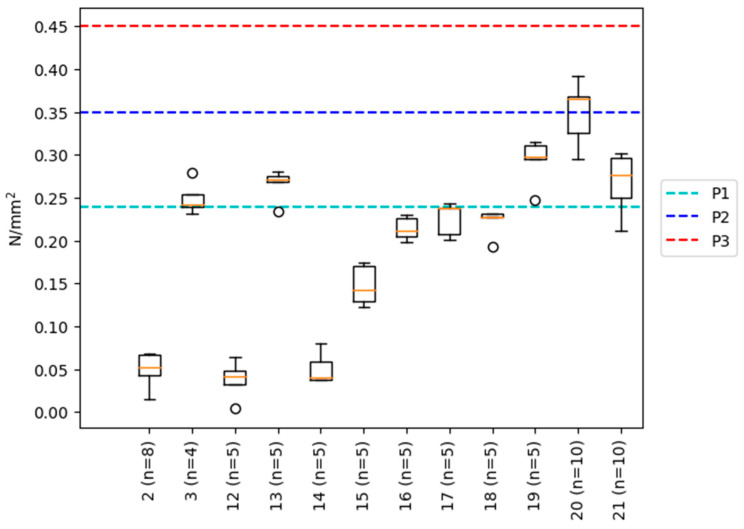
Boxplot of the IB results for all tested particleboards. P1, P2 and P3 refer to the 5% percentile IB value given by SN EN 312 for the first three particleboard classes (P1–P3). The n refers to the number of IB samples tested from the produced board. White circles correspond to outliers.

**Table 1 polymers-15-02668-t001:** Aliphatic and phenolic content of the lignin source 1 and source 2 using ^13^C-NMR [25].

Kraft Lignin	Aliphatic OH mmol/g	Phenolic OH mmol/g
S1	2.72	4.49
S2	2.66	4.45

**Table 2 polymers-15-02668-t002:** Gel time at 100 °C of the adhesive with varying 5-HMF/lignin S1 molar ratio.

5-HMF/Lignin S1 Molar Ratio	Gel Time (min)	pH	Solid Content
0.7	16	10.8	34.1
1	9	10.5	40.9
1.5	11	10.2	44.2
2	18	10.0	49.5
3	42	9.6	56.1

## Data Availability

The data presented in this study is available upon request from the corresponding author.

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
