# Peer review of "Exploring Fully Biobased Adhesives: Sustainable Kraft Lignin and 5-HMF Adhesive for Particleboards"

_polymers, 2023, doi:10.3390/polym15122668_

Round 1
Reviewer 1 Report
The manuscript entitled “ Exploring fully biobased adhesives: Sustainable Kraft lignin 2 and 5-HMF adhesive for particleboards” focused on the development of an ecological resin based on kraft lignin and 5-HMF for use in the production of particleboards. The research concept is well-prepared and of practical interest, the paper is well-written and clear. The reviewer would like to recommend the manuscript for publication after addressing the following points:
1. P3 L125 - What temperature was the resin cooled to?
2. P4 L149 - Further in the results authors state the amount of added resin for the production of particleboards, but I think it would be appropriate to state the value here in the methodology as well.
3. P5 L204 - For the value 1.5, the abbreviation for the hour is missing.
4. P7 L239 - In Figure 3, descriptions B and C are switched
5. P7 L261 - There is a typing error after the word Furthermore
6. P15 - In Appendix C, I would recommend to list the units for Press Closing Time and Press factor
Author Response
Thank you very much for your review and recommendation for publication.
The points were addressed as follows in the reviewed manuscript.
- P3 L125: The information regarding what temperature the resin was cooled to was added. It was room temperature.
- P4 L149: Section 2.5 was updated with this and further information.
- P5 L204: The hour abbreviation was added.
- P7 L239: The description in Figure 3 was corrected.
- P7 L261: The typing error was corrected by replacing the full stop with a comma and then correcting the capitalisation of “it”.
- P15: The units have been added to the table.
Hopefully, this clears up any remaining issues; if not, we can undertake further corrections.
Regards,
Arthur Thirion
Reviewer 2 Report
This manuscript deals with the use of biobased adhesives for the preparation of particleboards. In particular, the replacement of phenol-formaldehyde resins by Kraft lignin for phenol substitution and 5- hydroxymethylfurfural (5-HMF) for formaldehyde substitution. The paper is very interesting, well written and the results obtained are very relevant. As the authors indicate the adhesive formulation has been patented. The only aspect to correct is to clarify which are the sources of lignin (described as 1 and 2). It is important to at least put the reference of the raw material if it has already been published in another paper or by patent. Perhaps the annexes could be published as additional results. After that the article can be published
Author Response
Thank you very much for your review and positive comments on the manuscript.
Regarding the lignin source, we have updated section 2.1 Materails with the two Kraft lignin suppliers. They were UPM and Stora Enso, respectively.
Additionally, we have opted to keep the annexes as is to keep the paper concise for readability. If this is an issue, we are amicable about changing our position.
Hopefully, this clears up any remaining issues. If not, we are available to undertake further corrections.
Regards,
Arthur Thirion
Reviewer 3 Report
Dear Authors,
I found your manuscript as of high potential interest to readers, due to the up-to-date topic you explored (non-formaldehyde wood binders).
Below, please find several remarks, which in my opinion should improve your manuscript before publication:
- lines 130, 133, 134 and others: please unify the pressure units
- line 137 - when saying about filler and extender ratio, it is given in weight? Please clarify this
- line 141 (Particleboard preparation description): please provide here all the data required to prepare your samples, including resination, maximum unit pressure, pressing time etc. This is the section to find these data and not the Results and Discussion section
- lines 167-168 - the information about the beech wood you used in ABES should be moved to the Materials section
- line 179 - you said you did measure the density profile; if so, please provide a broader/more detailed description of how you did this and on what equipment, and, please provide sufficient results in the proper section; also, the dimensions you provided (50 mm x 50 mm) are given in the standard you used, so these can be removed
- line 218 (Fig. 2) and other - please use a dot instead of a comma in decimals
- lines 299-304 - please move it to the proper section in Methodology
Best regards!
Author Response
Thank you very much for your review and positive comments on the manuscript.
We have acted upon your remarks in the following manner:
- Lines 130, 133 and 134: The units for pressure were unified to bars across the paper. Furthermore, minutes were consistently abbreviated to min. MPa were switched to N/mm^2. Weight percentage units were further harmonised to wt%.
- Lines 137: We have clarified the filler and extender ratio in the text. It was based on weight.
- Lines 141 and 299-304: We have updated section 2.5 particleboard preparation with all the requested information and moved the relevant info from the results and discussion to methodology.
- Lines 167-168: The information about the beech wood veneers has been moved to the materials section 2.1.
- Lines 179: The information on the density profile measurement has been removed as we have elected not to include density profiles in the paper as results indicate issues with the methodology and was kept for a future investigation.
- Fig 2,3,4 and 5: We have switched to dots for the numbers in the different graphs.
Hopefully, this clears up any remaining issues. If not, we are available to undertake further corrections.
Regards,
Arthur Thirion